

# T1000: a reduced gene set prioritized for toxicogenomic studies

Othman Soufan[1], Jessica Ewald[2], Charles Viau[1], Doug Crump[3], Markus Hecker[4], Niladri Basu[2] and Jianguo Xia[1,5]

[1] Institute of Parasitology, McGill University, Montreal, Canada
[2] Faculty of Agricultural and Environmental Sciences, McGill University, Montreal, Canada
[3] Ecotoxicology and Wildlife Health Division, Environment and Climate Change Canada, National Wildlife Research Centre, Carleton University, Ottawa, Canada
[4] School of the Environment & Sustainability and Toxicology Centre, University of Saskatchewan, Saskatoon, Canada
[5] Department of Animal Science, McGill University, Montreal, Canada

Corresponding authors
Niladri Basu, niladri.basu@mcgill.ca
Jianguo Xia, jeff.xia@mcgill.ca

## ABSTRACT

There is growing interest within regulatory agencies and toxicological research communities to develop, test, and apply new approaches, such as toxicogenomics, to more efficiently evaluate chemical hazards. Given the complexity of analyzing thousands of genes simultaneously, there is a need to identify reduced gene sets. Though several gene sets have been defined for toxicological applications, few of these were purposefully derived using toxicogenomics data. Here, we developed and applied a systematic approach to identify 1,000 genes (called Toxicogenomics-1000 or T1000) highly responsive to chemical exposures. First, a co-expression network of 11,210 genes was built by leveraging microarray data from the Open TG-GATEs program. This network was then re-weighted based on prior knowledge of their biological (KEGG, MSigDB) and toxicological (CTD) relevance. Finally, weighted correlation network analysis was applied to identify 258 gene clusters. T1000 was defined by selecting genes from each cluster that were most associated with outcome measures. For model evaluation, we compared the performance of T1000 to that of other gene sets (L1000, S1500, Genes selected by Limma, and random set) using two external datasets based on the rat model. Additionally, a smaller (T384) and a larger version (T1500) of T1000 were used for dose-response modeling to test the effect of gene set size. Our findings demonstrated that the T1000 gene set is predictive of apical outcomes across a range of conditions (e.g., *in vitro* and *in vivo*, dose-response, multiple species, tissues, and chemicals), and generally performs as well, or better than other gene sets available.

## INTRODUCTION

Over the past decade there have been profound steps taken across the toxicological sciences and regulatory communities to help transform conventional toxicity testing largely based on animal models and apical outcome measurements to an approach that is founded on systems biology and predictive science (*Kavlock et al., 2018*; *Knudsen et al., 2015*; *Villeneuve*

*& Garcia-Reyero, 2011*). On the scientific side, efforts are being exemplified by emergent notions such as the Adverse Outcome Pathway framework (AOP; *Ankley et al., 2010*) and New Approach Methods (*ECHA, 2016*). On the regulatory side, these are exemplified by changes to, for example, chemical management plans in Canada, the United States and REACH (*ECHA, 2007*) across the European Union.

A core tenet underlying the aforementioned transformations, as catalyzed by the 2007 U.S. National Research Council report "Toxicity Testing in the 21st Century" (*NRC, 2007*), is that perturbations at the molecular-level can be predictive of those at the whole organism-level. Though whole transcriptome profiling is increasingly popular, it still remains costly for routine research and regulatory applications. Additionally, building predictive models with thousands of features introduces problems due to the high dimensionality of the data and so considering a smaller number of genes has the potential to increase classification performance (*Alshahrani et al., 2017*; *Soufan et al., 2015b*). Identifying smaller panels of key genes that can be measured, analyzed and interpreted conveniently remain an appealing option for toxicological studies and decision making

In recent years, several initiatives across the life sciences have started to identify reduced gene sets from whole transcriptomic studies. For example, the Library of Integrated Network-Based Cellular Signatures (LINCS) project derived L1000, which is a gene set of 978 'landmark' genes chosen to infer the expression of 12,031 other highly connected genes in the human transcriptome (*Subramanian et al., 2017*). In the toxicological sciences, the US Tox21 Program recently published S1500+, which is a set of 2,753 genes designed to be both representative of the whole-transcriptome, while maintaining a minimum coverage of all biological pathways in the Kyoto Encyclopedia of Genes and Genomes (KEGG) database (*Kanehisa et al., 2007*) and the Molecular Signatures Database (MSigDB) (*Liberzon et al., 2015a*). The first 1,500 genes were selected by analyzing microarray data from 3,339 different studies, and the rest were nominated by members of the scientific community (*Mav et al., 2018*). L1000 and S1500 gene sets were originally proposed to serve a different purpose. The 978 landmark genes of L1000 are chosen to infer expression of other genes more accurately, while genes of S1500 are selected to achieve more biological pathway coverage. Compared to L1000, the S1500 gene set attains more toxicological relevance through the gene nomination phase, though its data-driven approach relies upon microarray data primarily derived from non-toxicological studies. It worth nothing that about 33.7% (i.e., intersection over union) of genes are shared between both signatures. Even though some differences can be realized between L1000 and S1500, they are both strong candidates of gene expression modeling and prediction (*Haider et al., 2018*).

The objectives of the current study were to develop and apply a systematic approach to identify highly-responsive genes from toxicogenomic studies, and from these to nominate a set of 1000 genes to form the basis for the T1000 (Toxicogenomics-1000) reference gene set. Co-expression network analysis is an established approach using pairwise correlation between genes and clustering methods to group genes with similar expression patterns (*van Dam et al., 2018*). First, a co-expression network was derived using *in vitro* and *in vivo* data from human and rat studies from the Toxicogenomics Project-Genomics Assisted Toxicity Evaluation System (Open TG-GATEs) database. Next, the connections within the

**Table 1** **Summary of datasets used in the current study.** Datasets 1–3 were used to develop T1000 (see Phase I, II & III in 'Methods') and datasets 4 and 5 (see Phase IV in 'Methods') were used to evaluate the performance of the gene sets.

| Dataset # | Dataset | Organism | Organ | Exposure type | Number of chemicals | Matrix size (% missing values) | Purpose in current study |
|---|---|---|---|---|---|---|---|
| 1 | Open TG-GATEs | Human | Liver | *in vitro* | 158 chemicals | 2,606 experiments × 20,502 genes (8.9%) | Training |
| 2 | Open TG-GATEs | Rat | Liver | *in vitro* | 145 chemicals | 3,371 experiments × 14,468 genes (11.6%) | Training |
| 3 | Open TG-GATEs | Rat | Liver | *in vivo* (single dose) | 158 chemicals | 857 experiments × 14,400 genes (11.5%) | Training |
| 4 | Open TG-GATEs | Rat | Kidney | *in vivo* (single dose) | 41 chemicals | 308 experiments × 14,400 genes (12.2%) | Testing |
| 5 | Dose–response (GSE45892) | Rat | Liver, Bladder, Thyroid | *in vivo* (repeated dose) | 6 chemicals | 30 experiments × 14,400 genes (0%) | Testing (external validation) |
| Total | | | | | | 7,172 experiments | |

co-expression network were adjusted to increase the focus on genes in KEGG pathways, the MSigDB, or the Comparative Toxicogenomics Database (CTD) (*Davis et al., 2017*). This incorporation of prior biological and toxicological knowledge was motivated by loose Bayesian inference to refine the computationally-prioritized transcriptomic space. Clusters of highly connected genes were identified from the resulting co-expression network, and machine learning models were applied to prioritize clusters based on their association with apical endpoints. Clustering genes based on expression data has been shown to be instrumental in functional annotation and sample classification (*Necsulea et al., 2014*), with the rationale that genes with similar expression patterns are likely to participate in the same biological pathways (*Budinska et al., 2013*). From each cluster, key genes were identified for inclusion in T1000. Testing and validation of T1000 was realized through two separate datasets (one from Open TG-GATEs and one from the US National Toxicology Program) that were not used for gene selection. The current study is part of the larger EcoToxChip project (*Basu et al., 2019*). For the processed data, users can download all samples processed from https://zenodo.org/record/3359047#.XUcTwpMzZ24. We also deposited source codes and scripts used for the study at https://github.com/ecotoxxplorer/t1000.

## MATERIALS & METHODS

### Databases and datasets preparation

The derivation of T1000 was based on five public microarray datasets of toxicological relevance (Table 1): four datasets from Open TG-GATEs (*Igarashi et al., 2014b*), and one dataset generated by Thomas et al. (referred to as the dose–response dataset in this manuscript; GSE45892) (*Thomas et al., 2013*). Table 1 provides a summary of all microarray

datasets used in this study. For building the initial T1000 gene set, we used three of the four Open TG-GATEs datasets (see datasets 1–3 in Table 1).

## Open TG-GATEs

Open TG-GATEs is one of the largest publicly accessible toxicogenomics resources (*Igarashi et al., 2014b*). This database comprises data for 170 compounds (mostly drugs) with the aim of improving and enhancing drug safety assessment. It contains gene expression profiles and traditional toxicological data derived from *in vivo* (rat) and *in vitro* (primary rat hepatocytes and primary human hepatocytes) studies. To process the raw gene expression data files of Open TG-GATEs, the Affy package (*Gautier et al., 2004*) was used to produce Robust Multi-array Average (RMA) probe set intensities (*Irizarry et al., 2003b*). Gene annotation for human and rat was performed using Affymetrix Human Genome U133 Plus 2.0 Array annotation data and Affymetrix Rat Genome 230 2.0 Array annotation data, respectively. Genes without annotation were excluded. When the same gene was mapped multiple times, the average value was used. Finally, all profiles for each type of experiment were joined into a single matrix for downstream analysis.

From the training datasets, specific samples were labelled binary as "dysregulated" or "non-dysregulated". Dysregulated refers to exposure cases with potential toxic outcomes and non-dysregulated included controls and exposures with non-toxic outcomes. For the *in vitro* datasets, gene expression changes were associated with lactate dehydrogenase (LDH) activity (%). The activity of LDH, which serves as a proxy for cellular injury or dysregulation, was binarized such that values above 105% and below 95% were considered "dysregulated". While conservative, we note that these cut-off values were situated around the 5% and 95% marks of the LDH distribution curve (see Fig. S1 and Supplemental Information 1 for more details).

For the *in vivo* datasets (kidney and liver datasets from Open TG-GATEs), gene expression changes were associated with histopathological measures. The magnitude of pathologies was previously annotated into an ordinal scale: present, minimal, slight, moderate and severe (*Igarashi et al., 2014a*). This scale was further reduced into a binary classification with the first three levels considered "non-dysregulated" while the latter two were considered "dysregulated".

## Dose–response dataset and benchmark dose (BMD) calculation

The dose–response dataset (Accession No. GSE45892), was used to externally evaluate the ability of T1000 genes to predict apical endpoints (*Thomas et al., 2013*). Briefly, this dataset contains Affymetrix HT Rat230 PM microarray data following *in vivo* exposure of rats to six chemicals (TRBZ: 1,2,4-tribromobenzene, BRBZ: bromobenzene, TTCP: 2,3,4,6-tetrachlorophenol, MDMB: 4,4′-methylenebis(*N*,*N*′-dimethyl)aniline, NDPA: N-nitrosodiphenylamine, and HZBZ: hydrazobenzene). In exposed animals, both gene expression and apical outcomes (liver: absolute liver weight, vacuolation, hypertrophy, microvesiculation, necrosis; thyroid: absolute thyroid weight, follicular cell hypertrophy, follicular cell hyperplasia; bladder: absolute bladder weight, increased mitosis, diffuse transitional epithelial hyperplasia, increased necrosis epithelial cell) were measured,

permitting the comparison of transcriptionally-derived benchmark doses ($BMD_t$) with traditional benchmark doses derived from apical outcomes (*Yang, Allen & Thomas, 2007*). The apical outcome-derived benchmark dose ($BMD_a$) for each treatment group was defined as the benchmark dose from the most sensitive apical outcome for the given chemical-duration group.

Raw gene expression data (CEL files) for the dose–response dataset were downloaded from GEO (Accession No. GSE45892), organized into chemical-exposure-duration treatment groups, and normalized using the RMA method (*Irizarry et al., 2003a*). Only expression measurements corresponding to genes in the T1000 gene (or T384 and T1500) set were retained, resulting in reduced gene expression matrices for each treatment group ($t = 24$). The reduced gene expression matrices were analyzed using BMDExpress 2.0 to calculate a toxicogenomic benchmark dose ($BMD_t$) for each treatment group (*Yang, Allen & Thomas, 2007*). Here, the $BMD_t$ was calculated as the dose that corresponded to a 10% increase in gene expression compared to the control (*Farmahin et al., 2017*). Within BMDExpress 2.0, genes were filtered using one-way ANOVA (FDR adjusted $p$-value cut-off = 0.05). A $BMD_t$ was calculated for each differentially expressed gene by curve fitting with exponential (degree 2–5), polynomial (degree 2-3), linear, power, and Hill models. For each gene, the model with the lowest Akaike information criterion (AIC) was used to derive the $BMD_t$.

The $BMD_t$s from individual genes were used to determine a treatment group-level $BMD_t$ using functional enrichment analysis with Reactome pathways (*Farmahin et al., 2017*). Note, we chose here to functionally enrich with Reactome since we utilized KEGG to derive the T1000 list. After functional enrichment analysis, significantly enriched pathways ($p$-value <0.05) were filtered such that only pathways with >3 genes and >5% of genes in the pathway were retained. The treatment group-level $BMD_t$ was calculated by considering the mean gene-level $BMD_t$ for each significantly enriched pathway and selecting the lowest value. If there were no significantly enriched pathways that passed all filters, no $BMD_t$ could be determined for that treatment group. The similarity of the $BMD_t$ to the benchmark dose derived from apical outcomes ($BMD_a$) was assessed by calculating the $BMD_t/BMD_a$ ratio and the correlation between $BMD_t$ and $BMD_a$ for all treatment groups (*Farmahin et al., 2017*). Following the same procedures, $BMD_t/BMD_a$ ratio and correlation statistics were determined from genes belonging to L1000, S1500, and Linear Models for Microarray Data (Limma) (*Smyth, 2005*) to provide a reference for the performance of T1000 genes.

## Databases for Computing Prior Knowledge

The CTD, KEGG, and Hallmark databases were mined to integrate existing toxicogenomics and broader biological knowledge into one network that represents the prior knowledge space. CTD is manually curated from the literature to serve as a public source for toxicogenomics information, currently including over 30.5 million chemical-gene, chemical-disease, and gene-disease interactions (*Davis et al., 2017*). Following the recommendations of *Hu et al. (2015)*, only "mechanistic/marker" associations were extracted from the CTD database, thus excluding "therapeutic" associations that are presumably less relevant to toxicology. The extracted subgraph contained 2,889 chemicals,

950 diseases annotated as toxic endpoints (e.g., neurotoxicity, cardiotoxicity, hepatotoxicity and nephrotoxicity), and 22,336 genes. KEGG pathways are a popular bioinformatics resource that help to link, organize, and interpret genomic information through the use of manually drawn networks describing the relationships between genes in specific biological processes (*Kanehisa et al., 2007*). The MSigDB Hallmark gene sets have been developed using a combination of automated approaches and expert curation to represent known biological pathways and processes while limiting redundancy (*Liberzon et al., 2015b*).

Each feature vector consisted of 239 dimensions, representing information encoded from Hallmark, KEGG and CTD. For the Hallmark and KEGG features, we used "1″or "0″to indicate if a gene was present or absent for each of the 50 Hallmark gene sets (*Liberzon et al., 2015b*) and 186 KEGG pathways (*Kanehisa & Goto, 2000*). These features were transformed into z-scores. For the CTD features, we computed the degree, betweenness centrality, and closeness centrality of each gene, based on the topology of the extracted CTD subgraph. The topology measures were log-scaled for each gene in the network. The resulting prior knowledge space consisted of a 239-dimension vector for each of the 22,336 genes, with each vector containing 50 z-score normalized Hallmark features, 186 z-score normalized KEGG features, and three log-scaled CTD network features.

### Reactome database

To understand the biological space covered by T1000, we analyzed T1000′s top enriched Reactome pathways (as KEGG was used to develop T1000). Reactome is a manually curated knowledgebase of human reactions and pathways with annotations of 7,088 protein-coding genes (*Croft et al., 2014*).

## Performance evaluation

For the performance evaluation and testing phase, we leveraged the fourth dataset from Open TG-GATEs (see dataset 4 in Table 1), which was not used for gene ranking or selection so that it could serve as an external validation dataset. The dose–response dataset was used for an additional external validation (see dataset 5 in Table 1).

In this step, we applied five supervised machine learning methods to the TG-GATES rat kidney *in vivo* dataset, with the objective to predict which exposures caused significant "dysregulation", according to the criteria defined in step 4. This dataset was purposefully not used earlier when deriving T1000 so that it could serve later as a validation and testing dataset. The five machine learning models used were K-nearest neighbors (KNN; $K = 3$) (*Cover & Hart, 1967*), Decision Trees (DT), Naïve Bayes Classifier (NBC), Quadratic Discriminant Analysis (QDA) and Random Forests (RF).

The performance of each method was evaluated with five-fold cross-validation and measured using six different metrics (Eqs. (1)–(6)). TP represents the number of true positives, FP the number of false positives, TN the number of true negatives and FN the number of false negatives. The $F_1$ score (also called the balanced F-score) is a performance evaluation measure that computes the weighted average of sensitivity and precision (*He & Garcia, 2009*), and is well-suited for binary classification models. The $F_{0.5}$ score (*Davis & Goadrich, 2006*; *Maitin-Shepard et al., 2010*; *Santoni, Hartley & Luban, 2010*) is

another summary metric that gives twice as much weight to precision than sensitivity. The evaluation was performed on a Linux based workstation with 16 cores and 64 GB RAM for processing the data and running the experiments.

$$sensitivity = TP/(TP + FN) \tag{1}$$

$$specificity = TN/(TN + FP) \tag{2}$$

$$precision = TP/(TP + FP) \tag{3}$$

$$GMean = \sqrt{sensitivity \times specificity} \tag{4}$$

$$F_1 Score = 2 \times \frac{precision \times sensitivity}{precision + sensitivity} \tag{5}$$

$$F_{0.5} Score = 1.25 \times \frac{precision \times sensitivity}{0.25 \times precision + sensitivity} \tag{6}$$

## Proposed T1000 Framework

The work of T1000 was conducted in four discrete phases as follows (see Fig. 1): (I) data preparation and gene co-expression network generation; (II) network clustering to group relevant genes; (III) gene selection and prioritization; and (IV) external testing and performance evaluation.

The goal of phase I was to construct two network representations of the interactions between toxicologically-relevant genes, with one based on TG-GATES microarray data (step 1&2) and the other based on the KEGG, MSigDB, and CTD databases (step 3). In a co-expression network, nodes represent genes and edges represent the Pearson's correlation of expression values of pairs of genes. In the current study, we constructed three separate co-expression networks using gene expression profiles from Open TG-GATEs datasets (human *in vitro*, rat *in vitro*, and rat *in vivo*) (Table 1). If an interaction with a correlation coefficient of 60% or higher was present in all three networks, that gene-gene interaction was then accepted and mapped into one integrated co-expression network by averaging the absolute values of the pairwise correlation coefficients between individual genes. Matching between rat and human genes was based on gene symbols (e.g., Ddr1 in rat is matched with DDR1 in human using BiomaRt R package *Durinck et al., 2009*) and ignored when no match exists. This is a more conservative approach to maintain perfect matching orthologues in the networks although other computational approaches to match orthologues can be used (*Wang et al., 2015*). The final integrated co-expression network had 11,210 genes from a total of 20,502 genes.
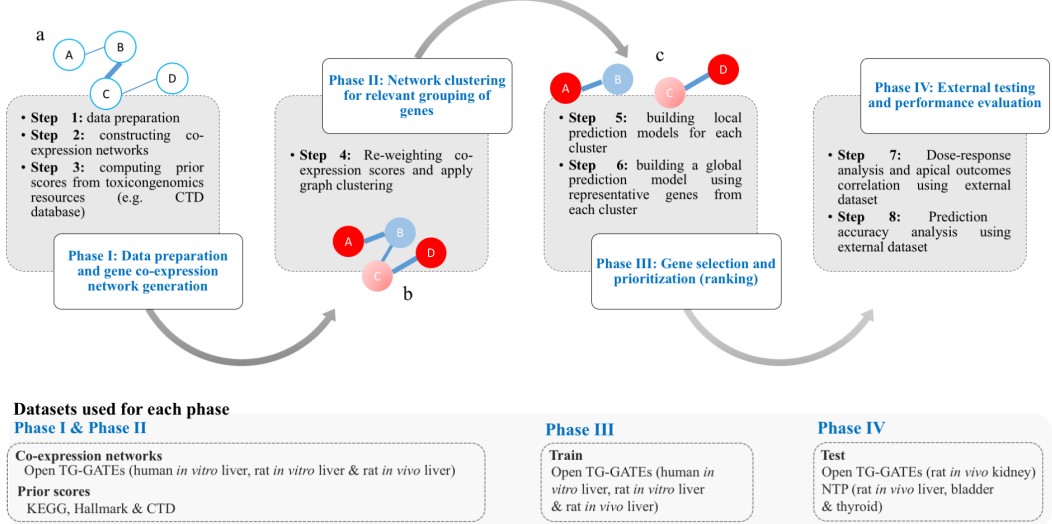

**Figure 1** **Framework of the T1000 approach for gene selection and prioritization.** Phase I is composed of Steps [1-3]. After data is prepared in Step 1, the co-expression network is generated through Step 2. The prior knowledge scores are computed using (KEGG, MSigDB) and toxicological (CTD) relevance graphs in Step 3. Phase II involves Step 4 for re-weighting of the co-expression scores based on prior knowledge of biological and toxicological relevance graphs. In addition, the graph is clustered during Step 4. In Phase III, in Step 5, a prediction model is trained for each cluster. Then, after selecting top genes from each cluster in Step 5, a one final prediction model called global is trained to rank all selected genes (Step 6). Phase IV is a focused on external evaluation of the prioritized gene list.

To build the prior knowledge space (step 3), we encoded information from the Hallmark, KEGG and CTD databases into feature vectors composed of 239 features describing each gene (see Materials section). Then, we projected the data onto a two-dimensional space using principle component analysis (PCA) and clustered using K-means ($K = 3$) to detect those genes that contributed most to the prior knowledge space. Regarding K-means, we initially experimented with $K = 1$, $K = 3$ and $K = 5$ and after visual inspection of summarized information as Supplemental Information 2, Fig. 1, we chose $K = 3$.

Genes that were furthest from the centroids (i.e., highest contributing ones) of the K-means clusters were more enriched with pathways and gene-chemical-disease interactions (see Supplemental Information S2). Based on step 3, a ranked list of all genes was generated such that the first ranked gene would have a prior score of 100% and the last, a prior score close to 0%. In phase II, we re-weighted the interactions in the co-expression network based on the prior knowledge space and then detected clusters of highly connected genes in the updated network (step 4). In a Bayesian fashion, the pairwise connections between genes in the co-expression network were re-weighted by multiplying the correlation with the mean prior score. For example, given $P(A)$ and $P(B)$ as prior scores of genes A and B, the correlation score $S(A, B)$ is re-weighted as follows (Eq. (7)):

$$S(A, B)_{new} = S(A, B) * ((P(A) + P(B))/2) \qquad (7)$$

It should be noted that in Eq. (7), the product of joint distribution could have been considered for the update such that $S(A,B)_{new} = S(A,B) * (P(A) * P(B))$.

After re-weighting the connections, we detected clusters of highly connected genes using the Markov Cluster Algorithm (MCL) (*Van Dongen & Abreu-Goodger, 2012*). The MCL approach groups together nodes with strong edge weights and then simulates a random flow through a network to find more related groups of genes based on the flow's intensity of movement. It does not require the number of clusters to be pre-specified. An inflation parameter controls the granularity of the output clustering and several values within a recommended range (1.2–5.0) were tried (*Van Dongen & Abreu-Goodger, 2012*). To optimize for the granularity of the clustering, a systematic analysis for the MCL inflation parameter was performed with values in range (1.2–5.0) (see Supplemental Information 3). After examining closely efficiency and mass fraction, a value of 3.3 was chosen. This generated 258 clusters that consisted of 11,210 genes. The average number of genes in each cluster was 43.4 with the min-max ranging from 1 to 8,423.

The goal of phase III of gene selection and prioritization was to select the top genes from each cluster to form T1000 (step 5), and then produce a final ranking of the 1,000 selected genes (step 6). For each of the 258 gene clusters, random forest (RF) classifiers were used to rank genes based on their ability to separate changes in gene expression labelled as "dysregulated" from those labelled "non-dysregulated", using the Gini impurity index of classification (*Nguyen, Wang & Nguyen, 2013*; *Qi, 2012*; *Tolosi & Lengauer, 2011*). RF is one of the most widely used solutions for feature ranking, and as an ensemble model, it is known for its stability (*Chan & Paelinckx, 2008*). In order to cover more biological space and ensure selected genes represent the whole transcriptome, a different RF classifier was built for each cluster and used to select representative genes (*Sahu & Mishra, 2012*).

We selected the top genes from each cluster based on the performance of the RF classifier. For example, when selecting the 1,000 top genes from two clusters (A and B), if the cross-validation prediction accuracy estimated for models A and B were 60% and 55%, respectively, then 522 ((60%/(60%+55%))*1000) and 478 ((55%/(60%+55%))*1000) genes would be selected from clusters A and B. However, if cluster A contained only 520 genes, the remaining two genes would be taken from group B, if possible. So, the cluster size is only used if it contains insufficient genes. We repeated this process until 1000 genes were selected. After choosing top *k* genes from each cluster, we aggregated them into a single list of 1000 genes and built a final RF model to get a global ranking of the genes. We refer to this final ranked list as T1000 (see Table S1 for a full list of selected genes and summary annotation; see Supplemental Information 4 for the cluster assignment of the genes). The goal of phase IV was to test the performance of the T1000 gene set using external datasets, and thus transition from gene selection activities to ones that focus on the evaluation of T1000. Phase IV is discussed in the following Results section. To discuss factors that characterize and distinguishe T1000 from L1000 and S1500, Table 2 is provided. As summarized in Table 2, T1000 is more toxicogenomic tailored by selecting genes that optimizes for endpoint predictions and using toxicogenomic datasets. Incorporating the prior knowledge space is critical for T1000 in ranking genes with more contribution to toxic effects. L1000 aims at finding a set of genes that can be used to extrapolate for the full

**Table 2 Descriptive comparison of T1000 against existing gene sets.** For the 'selection criteria' column, expression space coverage refers to the goal of finding a subset of genes that would achieve high correlation with the original full set of genes. Pathway coverage refers to finding a subset of genes that cover more pathways in a reference library.

| Gene set | Selection criteria | Ranked gene list | Species | Data | Approach | Number of genes |
|---|---|---|---|---|---|---|
| L1000 | Expression space coverage | No | Human | L1000 data | PCA and clustering (Data mining) | 978 |
| S1500 (NTP 2018) | Pathway coverage that combines data-driven and knowledge-driven activities | No | Human | Public GEO expression datasets (mainly GEO 3339 gene expression series) | PCA, clustering, and other data-driven steps (Data mining) | 2,861 (includes L1000 genes) |
| T1000 | Toxicological relevance using endpoint prediction | Yes | Human and Rat | Open TG-GATEs that is founded on co-expression networks from CTD, KEGG and Hallmark | Co-expression network and prior knowledge (Graph mining). PCA and clustering are used only for the prior knowledge. | 1,000 |

expression space of all other genes. S1500 has considered an optimization for the number of covered pathways. T1000, L1000 and S1500 have considered using PCA and clustering during the selection process. In T1000, however, this step is part of computing the prior only.

# RESULTS

## Overview of T1000 and biological relevance

The genes comprising T1000 cover a wide biological space of toxicological relevance. For illustration, co-expression networks, before and after applying Steps 2 and 3 (i.e., networks built on the Open TG-GATEs data that are subsequently updated with prior information from KEGG, MSigDb, and CTD), are shown in Fig. 2. In Fig. 2A, a sample co-expression network composed of 150 genes (i.e., 150 for visualization purposes only; of the 11,210 genes identified) has, in general, similar color and size of all the nodes of the network. While this covers a broad toxicological space, it does not necessarily identify or prioritize the most important genes. After subjecting the data to steps 2 and 3, two clusters of genes with different node sizes and colors were identified (Fig. 2B). Through this refined network, we then applied a prediction model to each cluster to identify the most representative genes resulting in the final co-expression network of the T1000 genes (Fig. 2C).

The complete list of T1000 genes with their gene symbols and descriptions, as well as their regulation states (up- or down-regulated) is provided in Table S1.

Visual examination of the Reactome enrichment map (Fig. S2) reveals that 'biological oxidations' (the largest circle in Fig. S2) contained the most enriched pathways followed by 'fatty acid metabolism'. This is logical given that xenobiotic and fatty acid metabolism, mediated by cytochrome P450 (CYP450) enzymes, feature prominently across the toxicological literature (*Guengerich, 2007*; *Hardwick, 2008*).

We further examine two genes that are ranked among the top up- and down-regulated gene sets, respectively. We observed that CXCL10 (ranked 2nd in up-regulated genes)

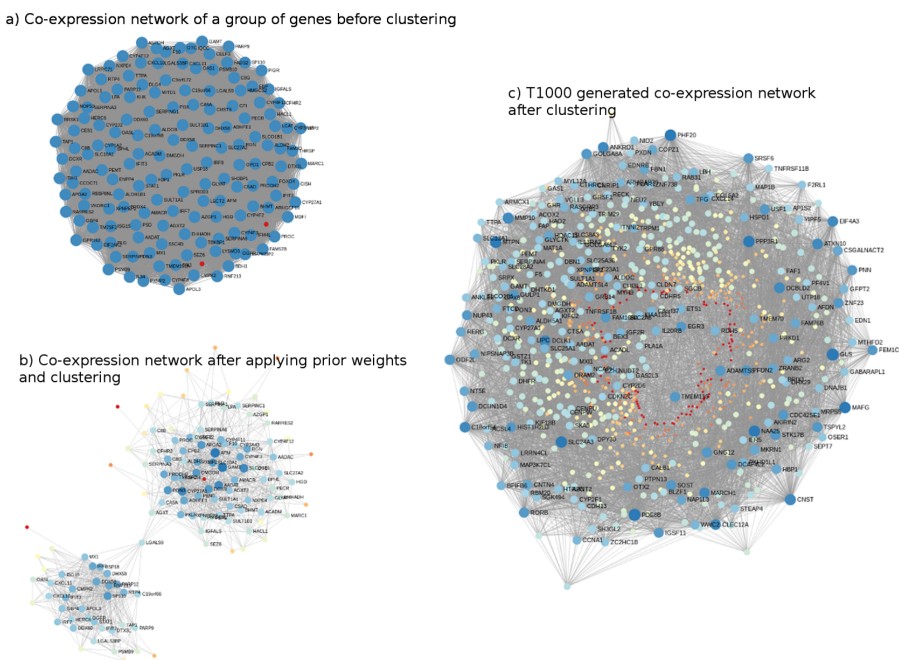

a) Co-expression network of a group of genes before clustering

c) T1000 generated co-expression network after clustering

b) Co-expression network after applying prior weights and clustering

**Figure 2** **Visual representation of co-expression networks before and after performing Steps 2 and 3 of the T1000 selection process.** Visual representation of co-expression networks before and after performing Steps 2 and 3 of the T1000 selection process. A sample co-expression network of a group of 150 genes such that each pair of genes would have a connection is provided in (A). After re-weighting the correlation scores using the prior knowledge of biological and toxicological relevance graphs and performing clustering through Steps [1-4] of T1000 framework (see Fig. 1), the graph in (A) is evolved to the one in (A). In (B), a pair of genes would have a link only if they hold enough confidence after applying prior scores. From (B), nodes representing genes gain different levels of colors summarizing different levels of structural representations in the graph. Therefore, it is more relevant to cluster the graph at this stage after applying prior weights instead of the stage of (A). We can visually detect two separate clusters of genes in (B). After executing T1000 framework, we visualize the generated co-expression graph of all selected 1,000 genes in (C). Compared to (A), we see variant levels of colors indicating different structural relevance. The colors in (A), (B), and (C) reflect structural statistics using betweenness centrality and node degree. (A) holds a very similar statistics while (B) and (C) exploits and shows variant levels. A more contributing gene would have a larger node and a darker blue color while a less important one would have a very small node with a red color intensity. Please note that (B) and (C) are realized only after executing steps from T1000 framework while (A) shows the generic representation of the co-expression graph.

and IGFALS (ranked 3rd in down-regulated genes) had reported links in the literature in response to exposure to toxic compounds. Upregulation of CXCL10, the ligand of the chemokine receptor CXCR3 found on macrophages, has been observed in the bronchiolar epithelium of patients with Chronic Obstructive Pulmonary Disease (COPD) compared to non-smokers or smokers with normal lung function (*Saetta et al., 2002*). Smokers develop COPD after exposure to the many chemicals found in cigarette smoke, which include oxidants that cause inflammation (*Foronjy & D'Armiento, 2006*). Although TG-GATEs does not contain any cigarette toxicants within its database, the general pathways by which toxicants disrupt tissue function are represented by T1000.

**Table 3  Summary of correlation of apical endpoints to 24 experimental groups (6 chemicals × 4 exposure durations).**

|  | T384 ($n = 384$) | T1000 ($n = 1,000$) | T1500 ($n = 1,500$) | L1000 ($n = 976$) | S1500 ($n = 2,861$) | Limma ($n = 1,000$) |
|---|---|---|---|---|---|---|
| # of $BMD_t$s | 18 | 21 | 21 | 21 | 21 | 14 |
| Mean ratio ($BMD_t/BMD_a$) | 2.2 | 1.2 | 1.1 | 1.8 | 1.1 | 2.1 |
| Correlation ($BMD_t$, $BMD_a$) | 0.83 ($p < 0.001$) | **0.89**($p < 0.001$) | 0.83($p < 0.001$) | 0.76($p < 0.001$) | 0.78($p < 0.001$) | 0.73($p < 0.01$) |

A gene that was found to be significantly downregulated by T1000 was the gene encoding for Insulin Like Growth Factor Binding Protein Acid Labile Subunit or IGFALS, which is an Insulin growth factor-1 (IGF-1) binding protein (*Amuzie & Pestka, 2010*). Interestingly, the mRNA expression of IGFALS was reported to be significantly downregulated when experimental animals were fed deoxynivalenol, a mycotoxin usually found in grain (*Amuzie & Pestka, 2010*). By reducing IGFALS, the half-life of circulating IGF-1 is reduced, causing growth retardation (*Amuzie & Pestka, 2010*). Many compounds in the TG-GATEs database are of organismal origin, and thus, as the data suggest, they have a similar mode of action as deoxynivalenol in reducing expression of important effectors such as IGFALS.

Regarding potential clinical applications, we discuss the use of T1000 signature for screening drugs that may show toxic adverse effects in Supplemental Information 5. The experiment is motivated by the connectivity map project for connecting small molecules, genes, and disease using gene-expression signatures (*Lamb et al., 2006*).

## Benchmark dose–response results

Overall, the aim of the evaluation was to assess the ability of T1000 gene sets to predict apical outcomes according to previously published methods (*Farmahin et al., 2017*). Additionally, we repeated step 4 of the T1000 approach to select the top 384 (T384; i.e., a number conducive to study in a QPCR microplate format as per the EcoToxChip project; *Basu et al., 2019*) and 1,500 (T1500 see Supplemental Information 6; i.e., a number pursued in other endeavours like S1500) genes to investigate the effect of gene set size on apical outcome prediction. To benchmark the performance of T1000 against other notable gene sets, we considered S1500 (*Merrick, Paules & Tice, 2015*) and L1000 (*Subramanian et al., 2017*).

$BMD_t$ analysis (see Materials section) of the dose–response dataset was performed with the T1000 gene list and the BMDExpress software program (*Yang, Allen & Thomas, 2007*). The maximum number of BMDs calculated was 21 because for three of the experimental groups a $BMD_a$ (benchmark dose, apical outcome) did not exist due to a lack of observed toxicity (Table 3). The T384 gene set performed similarly with Limma; however, increasing the size of this gene set to T1000 resulted in performance evaluation metrics that rivaled that of all other gene sets of the same size or larger (L1000, Limma, and S1500). Further increasing the size of T1000 to T1500 did not increase the performance as the correlation slightly decreased while the average ratio of $BMD_t/BMD_a$ got slightly closer to one. Figure 3 provides a visual summary of the comparison based on the $BMD_t/BMD_a$ ratios.

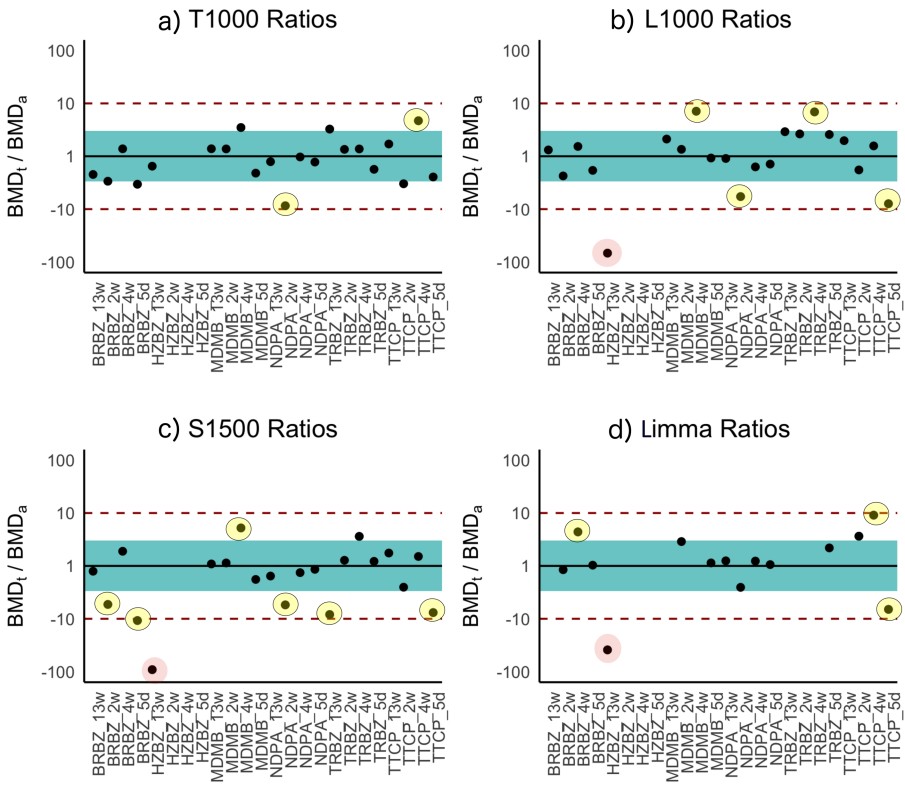

**Figure 3** **Ratios of BMDt/BMDa for each experimental group determined with various gene sets as indicated atop the plots.** Ratios of BMDt/BMDa represents ratio of transcriptionally-derived benchmark doses BMDt using gene signatures to apical outcome-derived benchmark dose BMDa serving as the ground truth. The limits of the blue rectangular band and dotted lines represent 3-fold and 10-fold of unity, respectively. Ratios could not be calculated for three experimental groups (hydrazobenzene (HZBZ): 5 day, 2 week, 4 week) due to a lack of apical outcomes. Red circles represent mean ratios greater than 10-fold, while the yellow ones represent ratios greater than 3-fold. The fewer circles, the more the gene set is indicative of potential relevance to the examined apical endpoints (see Figs. S3 and S4 for T384 and T1500 plots, respectively). In (A), the T1000 results are highlighted such that in only two experiments, the ratio of difference from the ground truth was greater than three folds and less than 10. In (A), (B), (C) and (D), the results of L1000, S1500 and Limma are illustrated, respectively, with each having a single experiment (i.e., red circle) with 10-fold difference from the ground truth. All of them had more yellow circles as compared to (A) of T1000.

## Prediction results

In a second validation study, we applied T1000 to study the Rat Genome 230 2.0 Array for the kidney dataset (dataset 4) from the Open TG-GATEs program. This dataset was not included in any model training or parameter tuning steps. This helped to establish another external validation of T1000 in terms of its generalized ability to predict apical outcomes for datasets derived from different tissues. When compared to the baseline gene sets mapped using Limma and L1000, T1000 achieved a relative improvement of the $F_1$ Score by 6.9% and 27.56%, respectively, thus outperforming the other gene sets (Table 4). When considering the absolute difference of $F_1$ Score between T1000 and the

**Table 4  Summary comparison of average classification performance using the testing RatKidney dataset.** Scores are based on average results from five classifiers (LDA, NBC, QDA, DT and RF) and the standard deviation is reported to highlight variance of estimate.

|  | Sensitivity | Specificity | Precision | Gmean | F1Measure | F0.5Measure |
|---|---|---|---|---|---|---|
| T1000 | **29.25% ($\pm$11.64)\*** | 71.33% ($\pm$4.74) | **21.51% ($\pm$4.45)** | **44.7% ($\pm$7.8)\*** | **24.58% ($\pm$7.11)\*** | **22.6% ($\pm$5.36)** |
| Limma | 27.76% ($\pm$16.3) | 70.75% ($\pm$6.33) | 20% ($\pm$9.96) | 41.84% ($\pm$14.81) | 22.99% ($\pm$12.04) | 21.06% ($\pm$10.64) |
| CD | 21.79% ($\pm$15.39) | 68.08% ($\pm$10.97) | 13.94% ($\pm$6.64) | 34.79% ($\pm$13.3) | 16.65% ($\pm$9.96) | 14.83% ($\pm$7.82) |
| L1000 | 22.99% ($\pm$12.82) | 70.42% ($\pm$5.78) | 16.84% ($\pm$7.29) | 38.33% ($\pm$11.46) | 19.27% ($\pm$9.27) | 17.71% ($\pm$7.97) |
| S1500 | 21.79% ($\pm$7.65) | **72.67% ($\pm$3.98)\*** | 17.87% ($\pm$3.99) | 39.19% ($\pm$6.2) | 19.53% ($\pm$5.42) | 18.48% ($\pm$4.48) |
| Random-500 | 27.83% ($\pm$11.69) | 70.89% ($\pm$5.09) | 20.31% ($\pm$4.89) | 42.81% ($\pm$8.38) | 18.41% ($\pm$12.03) | 21.29% ($\pm$5.79) |
| P-value (T1000 vs. Random) | 0.0555 | 0.3454 | 0.1283 | 0.0504 | 0.0192 | 0.1112 |
| Best Model (Limma_NBC) | 44.78% | 68.75% | 28.57% | 55.48% | 34.88% | 30.80% |
| Worst Model (Limma_QDA) | 4.48% | 72.08% | 4.29% | 17.97% | 4.38% | 4.32% |

**Notes.**
  \*Statistically significant at an alpha level of 0.1 using $T$-test and considering comparison with Random results.

second best (i.e., Limma), T1000 achieved an improvement of 1.59%. The improvement was 1.54% for $F_{0.5}$ Score confirming that T1000 led to fewer false positive predictions.

Another baseline we compare with is Random-500, where a set of 1000 features are selected randomly and the performance is reported for the five classifiers considered (i.e., LDA, NBC, KNN, QDA and RF). This experiment is repeated for 500 times and the average and standard deviation scores are reported in Table 4. GMean, $F_1$ Score and $F_{0.5}$ Score of T1000 are significantly higher ($t$-test with alpha = 0.1) than the random scores. The $t$-test we performed was based on the average performance of the five used different machine learning classifiers. So, we averaged results of Random-500 to get a summary performance scores for each of the classifiers. One observation is that the Random-500 results outperformed several gene sets. This can be due to the fact that some machine learning models are less sensitive to the type of selected features (e.g., RF). On average, we found that a randomly generated set would outperform other models with a chance of about 30% only. Here, we focused on $F_{0.5}$ Measure as one of the summary performance measures. It should be noted that this does not reflect the magnitude of improvement which is measured using the $t$-test. Given the fact that other approaches will outperform a random selection in 70% and with a significantly higher performance on average (see T1000 in Table 4), we conclude that a systematic approach is required to prioritize genes. In the context of high throughput screening, such small improvements in $F_1$ Score or $F_{0.5}$ Score may represent large cost savings (*Soufan et al., 2015a*) as false positives may lead to added experiments that would otherwise be unnecessary. Detailed performance scores of each individual machine learning model are provided in Table S2. Please refer to Supplemental Information 7 for more comparisons including expression space visualization using PCA and gene set coverage evaluation.

## DISCUSSION

There is great interest across the toxicological and regulatory communities in harnessing transcriptomics data to guide and inform decision-making (*Basu et al., 2019*; *Council, 2007*; *ECHA, 2016*; *Mav et al., 2018*; *Thomas et al., 2019*). In particular, gene expression signatures hold great promise to identify chemical-specific response patterns, prioritize chemicals of concern, and predict quantitatively adverse outcomes of regulatory concern, in a cost-effective manner. However, the inclusion of full transcriptomic studies into standard research studies faces logistical barriers and bioinformatics challenges, and thus, there is interest in the derivation and use of reduced but equally meaningful gene sets.

Our approach to select T1000 followed the same rationale of how the LINCS program derived the L1000 gene set (*Liu et al., 2015*), though here we purposefully included additional steps to bolster the toxicological relevance of the resulting gene set. Generating a list of ranked genes based on toxicologically relevant input data and prior knowledge is another key feature of T1000.

There are some limitations associated with our current study. For instance, the co-expression network was based on data from the Open TG-GATEs program. While this is arguably the largest toxicogenomics resource available freely, the program is founded on one *in vivo* model (rat), two *in vitro* models (primary rat and human hepatocytes), 170 chemicals that are largely drugs, and microarray platforms. Thus, there remain questions about within- and cross- species and cell type differences, the environmental relevance of the tested chemicals, and the biological space captured by the microarray. Our multi-pronged and -tiered bioinformatics approach was designed to yield a toxicologically robust gene set, and the approach can be ported to other efforts that are starting to realize large toxicogenomics databases such as our own EcoToxChip project (*Basu et al., 2019*). In addition, our approach in selecting T1000 genes was purely data-driven without considering input from scientific experts as was done by the NTP to derive the S1500 gene set (*Mav et al., 2018*). It is unclear how such gene sets (e.g., T1000, S1500) will be used by the community and under which domains of applicability, and thus there is a need to perform case studies in which new methods are compared to traditional methods (*Kavlock et al., 2018*). It is worth mentioning that T1000 had 259 and 90 genes in common with S1500 and L1000, respectively and 741 unique genes.

## CONCLUSIONS

Here we outlined a systematic, data-driven approach to identify highly-responsive genes from toxicogenomics studies. From this, we prioritized a list of 1,000 genes termed the T1000 gene set. We demonstrated the applicability of T1000 to 7,172 expression profiles, showing great promise in future applications of this gene set to toxicological evaluations. We externally validated T1000 against two *in vivo* datasets of toxicological prominence (a kidney dataset of 308 experiments on 41 chemicals from Open TG-GATEs and a dose–response study of 30 experiments on six chemicals (*Thomas et al., 2013*). We compared the performance of T1000 against existing gene sets (Limma, L1000 and S1500) as well as panels of randomly selected genes. In doing so, we demonstrate T1000′s versatility as

it is predictive of apical outcomes across a range of conditions (e.g., *in vitro* and *in vivo*), and generally performs as well as or better than other gene sets available. Our approach represents a promising start to yield a toxicologically-relevant gene set. We hope that future efforts will start to use and apply T1000 in a diverse range of settings, and from these we can then start to make updates to the composition of the T1000 gene set based on improved understanding of its performance characteristics and user experiences.

## ACKNOWLEDGEMENTS

We acknowledge the support of all members of the EcoToxChip project. We are grateful to the guidance offered by our project's program officer (Micheline Ayoub, Génome Québec) and members of our Research Oversight Committee (Chair: Nancy Denslow; Members: Kevin Crofton, Dan Schlenk, Roy Suddaby, and Carole Yauk).

### Funding

This study was funded by Genome Canada, Génome Québec, Genome Prairie, the Government of Canada, Environment and Climate Change Canada, Ministère de l'Économie, de la Science et de l'Innovation du Québec, the University of Saskatchewan, and McGill University. The funders had no role in study design, data collection and analysis, decision to publish, or preparation of the manuscript.

### Grant Disclosures

The following grant information was disclosed by the authors:
Genome Canada.
Génome Québec.
Genome Prairie.
the Government of Canada.
Environment and Climate Change Canada.
Ministère de l'Éco nomie, de laScienceet de l'Innovation du Québec.
the University of Saskatchewan.
McGill University.

### Competing Interests

Jianguo Xia is an Academic Editor for PeerJ.

### Author Contributions

- Othman Soufan and Jessica Ewald conceived and designed the experiments, performed the experiments, analyzed the data, contributed reagents/materials/analysis tools, prepared figures and/or tables, authored or reviewed drafts of the paper, approved the final draft.
- Charles Viau conceived and designed the experiments, analyzed the data, contributed reagents/materials/analysis tools, prepared figures and/or tables, authored or reviewed drafts of the paper, approved the final draft, manuscript review.

- Doug Crump and Markus Hecker conceived and designed the experiments, analyzed the data, contributed reagents/materials/analysis tools, authored or reviewed drafts of the paper, approved the final draft, manuscript review.
- Niladri Basu and Jianguo Xia conceived and designed the experiments, analyzed the data, contributed reagents/materials/analysis tools, authored or reviewed drafts of the paper, approved the final draft.

## Data Availability

The data is available at Open TG-GATEs and GEO: GSE45892.

Our processed version of Open TG-GATEs is available at Othman Soufan. (2019). Datasets used for T1000 [Data set]. Zenodo. http://doi.org/10.5281/zenodo.3359047.

## Supplemental Information

Supplemental information for this article can be found online at http://dx.doi.org/10.7717/peerj.7975#supplemental-information.

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
