# Peer review of "T1000: a reduced gene set prioritized for toxicogenomic studies"

_PeerJ, doi:10.7717/peerj.7975_

## Round 0.1 · original submission · Major Revisions

The manuscript got critical remarks demanding major revision. Please mention in the abstract that the comparisons were done on rat model, not on human. Experimental design presentation should be updated.

Reviewer 1 ·

Basic reporting

In this manuscript, Soufan et al. used a systematic approach to identify 1000 genes (T1000) highly responsive to chemical exposure using microarray data from the Open TG-GATEs program. The authors then compared the performance of T1000 to that of previously published gene sets (L1000, S1500, gene list from Limma, and a random set).
The manuscript is generally well written, and provides adequate background/literature review supporting the need of further studies aiming at building models to predict toxicity. Yet some major concerns remain.

1. When referring to the datasets used in the study, GEO Accession numbers should be provided in the main text (i.e. lines 134-137) as well as included in table 1.

2. The figure legends should be more exhaustive, to allow the reader to understand the main content of all the figures/tables without coming back to the main text.

3. Table 2 has not been discussed in the manuscript.

4. In the legend for figure 4, the abbreviations used in the graphs should be explained (i.e. HZBZ).

5. Raw and processed data, as well as source codes should be made available in accordance with the journal’s Data Sharing policy.

Experimental design

The method section is quite fragmented and sometimes repetitive. The workflow should be described in the “overview” and not repeated afterwards. Nonetheless, the authors provide a clear and detailed description of the analyses performed.

Validity of the findings

1. As the authors do not discuss how (or even in which contexts) their findings are going to guide decision making, the title of the manuscript “T1000: A reduced toxicogenomics gene set for improved decision making” should be rephrased. Indeed the authors fail to identify a clear use of their gene set for the scientific community (lines 444-447), thus bringing into question the value/scientific impact of the work.

2. The authors claim that T1000 has more toxicological relevance than other available gene sets (L1000, S1500) as their data-driven approach relies upon data derived from in vitro and in vivo human and rat toxicological studies. Yet table 4 shows that the sensitivity, specificity and precision of all of these gene sets are (a) very similar to a randomly selected gene set and (b) not impressive as classifiers (~25% sensitivity, ~70% specificity). This would lead to a lot of false negatives and a moderate number of false positives, bringing into question the value of T1000 and other gene sets for the identification of chemical hazards. Of further concern is the fact that these classifier values relate to the prediction of apical outcomes in a rat in vivo data set, in which histopathology data would also be available. The more useful scenario in which gene sets could be employed is to predict the clinical translation of a chemical hazard identified in preclinical studies. It would be expected that the classifier performance values would be even worse in this setting, hence the value of such an approach is questionable. Evidence supporting the translatability of T1000 from non-clinical species to human should therefore be provided. How well does T1000 perform as a classifier of drugs with e.g. hepatotoxic concern in humans?

3. The authors do not discuss their findings in a biological context. For instance, are there differences in genes/network responsible for the better performance of T1000 compared to the other available gene sets? This should be discussed in more detail in the result section (lines 365-386) and in the discussion.

4. Other authors have used the TG-GATEs databases to apply a weighted gene co-expression network analysis to safety assessment, identifying marker genes of liver toxicity but also using co-expression modules to understand stress-response pathways in drug-induced liver toxicity and to reveal mechanisms of pathogenesis concurrent with or preceding toxicity phenotypes (Sutherland JJ, et al. Toxicogenomic module associations with pathogenesis: a network-based approach to understanding drug toxicity. Pharmacogenomics J. 2018 May 22;18(3):377-390). Without any attempt to demonstrate the utility of T1000 for clinical applications, the paper lacks novelty in the biological/clinical sciences, and the authors may want to consider submitting to a journal in a different field (e.g. bioinformatic/mathematical modelling).

Reviewer 2 ·

Basic reporting

No comment

Experimental design

No comment

Validity of the findings

No comment

Additional comments

Dear Author

I have only two comments.

1) Reactome is not mentioned in Material and Methods.

2)The Authors addressed that the gene set was derived from analyzing data from human cell lines and in vivo and in vitro Rat liver. However, the testing and comparison with other known gene sets were done in rat tissues. Regarding the nature of how the authors calculated the gene set (involving both species and several tissues), the authors should mention in the Abstract that the testing was done in rat kidney data (or the dataset if they will)

Reviewer 3 ·

Basic reporting

no comment

Experimental design

1- Method seems to be fragmented, not well describe and hard to reproduce the results using the provided description. For instance:
a. Authors mentioned that they combined rat and human networks, however, they did not provide details on how the genes were mapped between the two species.
b. Some parameters are just provided without any justification, for example, authors set k for K-means clustering for 3 and MCL inflation parameter to 3.3. To obtain systematic results, authors should apply the proper hyperparameter selection for algorithms used in this study and share their selection criteria.
c. Authors refer to outliers in their Kmeans clustering method as most contributing, however, it is not clear to what these points contribute.
2- In the prior knowledge part authors, authors used distance from cluster centroids to rank the gene. This may work only if the cluster sizes are comparable, however, if the cluster sizes are different then the authors may need to normalize such distance.
3- Authors used the average of prior knowledge scores, this does not reflect the joint distribution of prior information of both nodes. Authors may need to explore further methods or discuss the limitations of their current method.
4- Authors used MCL to cluster their graph, MCL is known to produce too many small clusters. From the number provided in the text, it seems there is one big cluster that covers 75% of genes while all other clusters are very small. It is interesting to see the distribution of the clusters size authors have. Also, I would suggest using the weighted gene coexpression network analysis (WGCNA) tool as it provides a utility to analyze co-expression networks.

Validity of the findings

5-From the results side, it is interesting to see how the T1000 gene set compare to other gene sets reported in the literature in term of the number of common and different genes.
6- From Figure 5, classification results seem very comparable. It is also very comparable to random gene set. Authors may consider performing some permutation test to make sure that the accuracy they are obtaining is significant from random genes.

Additional comments

7- References to some of the figures are missing, lines 375, 384-386.
8- Figures 2 and 3 are hard to follow and see. I would suggest moving them to supplementary.

---

## Round 0.2 · Minor Revisions

Two reviewers have no more critical comments. But I still have concern regarding statistical significance of the model. Please update the text based on remarks by third reviewer on statistics. As academic editor I believe this paper does not need another reviewing round. Waiting the manuscript update with your comments on statistics.

Reviewer 1 ·

Basic reporting

The authors addressed all of the comments in my first review.

Experimental design

No comment

Validity of the findings

No comment

Additional comments

The authors addressed all of the comments in my first review.

Reviewer 2 ·

Basic reporting

No comment

Experimental design

No comment

Validity of the findings

No comment

Reviewer 3 ·

Basic reporting

Authors have substantially improved their manuscript, however there are still key issue related with randomness of T1000 genes.

Experimental design

1- Authors did not explain how did they measure the statistical significance of their model performance given the random-500 null hypothesis.

Validity of the findings

2- Author report significant level of 0.1 however on the other hand they say that models from random genes outperformed their model 30% of the times. Both results do not add up. In significance analysis 30% is considered a high number and indicate that the generated models are more likely random than true.

---

## Round 0.3 · accepted · Accept

As I see all the remaining remarks were taken into account in the updated manuscript, and commented in the rebuttal letter. Two reviewers recommended accept it even at previous reviewing round. As editor I think this paper may be published in current form.